# Unsupervised Audiovisual Synthesis via Exemplar Autoencoders

**Kangle Deng, Aayush Bansal, Deva Ramanan**
Carnegie Mellon University
Pittsburgh, PA 15213, USA
{kangled,aayushb,deva}@cs.cmu.edu

## Abstract

We present an unsupervised approach that converts the input speech of any individual into audiovisual streams of potentially-infinitely many output speakers. Our approach builds on simple autoencoders that project out-of-sample data onto the distribution of the training set. We use exemplar autoencoders to learn the voice, stylistic prosody, and visual appearance of a specific target exemplar speech. In contrast to existing methods, the proposed approach can be easily extended to an arbitrarily large number of speakers and styles using only 3 minutes of target audio-video data, without requiring *any* training data for the input speaker. To do so, we learn audiovisual bottleneck representations that capture the structured linguistic content of speech. We outperform prior approaches on both audio and video synthesis. Please visit our project website[1] for our summary video and more information.

## 1 Introduction

We present an unsupervised approach to retargeting the speech of any unknown speaker to an audiovisual stream of a known target speaker. Using our approach, one can retarget a celebrity video clip to say the words "Welcome to ICLR 2021" in different languages including English, Hindi, and Mandarin (please see our associated video). Our approach enables a variety of novel applications because it eliminates the need for training on large datasets; instead, it is trained with unsupervised learning on only a few minutes of the *target* speech, and does not require any training examples of the input speaker. By retargeting input speech generated by medical devices such as electrolarynxs and text-to-speech (TTS) systems, our approach enables personalized voice generation for voice-impaired individuals (Kain et al., 2007; Nakamura et al., 2012). Our work also enables applications in education and entertainment; one can create interactive documentaries about historical figures in their voice, or generate the sound of actors who are no longer able to perform. We highlight such representative applications in Figure 1.

**Prior work** typically independently looks at the problem of audio conversion (Chou et al., 2019; Kaneko et al., 2019a;b; Mohammadi & Kim, 2019; Qian et al., 2019) and video generation from audio signals (Yehia et al., 2002; Chung et al., 2017; Suwajanakorn et al., 2017; Zhou et al., 2019; Zhu et al., 2018). Particularly relevant are zero-shot audio translation approaches (Chou et al., 2019; Mohammadi & Kim, 2019; Polyak & Wolf, 2019; Qian et al., 2019) that learn a *generic* low-dimensional embedding (from a training set) that are designed to be agnostic to speaker identity (Fig. 2-a). We will empirically show that such generic embeddings may struggle to capture stylistic details of in-the-wild speech that differs from the training set. Alternatively, one can directly learn an audio translation engine *specialized* to specific input and output speakers, often requiring data of the two speakers either aligned/paired (Chen et al., 2014; Nakashika et al., 2014; Sun et al., 2015; Toda et al., 2007) or unaligned/unpaired (Chou et al., 2018; Fang et al., 2018; Kameoka et al., 2018; Kaneko & Kameoka, 2017; Kaneko et al., 2019a;b; Serrà et al., 2019). This requirement restricts such methods to known input speakers at test time (Fig. 2-b). In terms of video synthesis from audio input, zero-shot facial synthesis approaches (Chung et al., 2017; Zhou et al., 2019; Zhu et al., 2018) animate the lips but struggle to capture realistic facial characteristics of the entire person.

---

[1] https://dunbar12138.github.io/projectpage/Audiovisual/

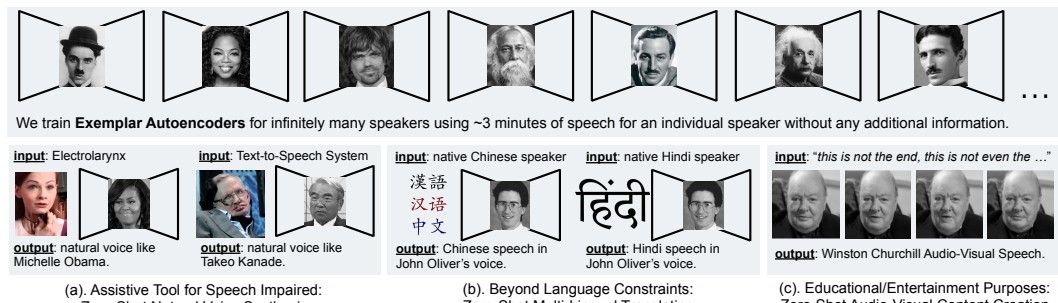

Figure 1: We train audiovisual (AV) exemplar autoencoders that capture personalized *in-the-wild* web speech as shown in the **top**-row. We then show three representative applications of Exemplar Autoencoders: (a) Our approach enables zero-shot natural voice synthesis from an Electrolarynx or a TTS used by a speech-impaired person; (b) Without any knowledge of Chinese and Hindi, our approach can generate Chinese and Hindi speech for John Oliver, an English-speaking late-night show host; and (c) We can generate audio-visual content for historical documents that could not be otherwise captured.

Other approaches (Ginosar et al., 2019; Shlizerman et al., 2018; Suwajanakorn et al., 2017) restrict themselves to known input speakers at test time and require large amounts of data to train a model in a supervised manner.

**Our work** combines the zero-shot nature of generic embeddings with the stylistic detail of person-specific translation systems. Simply put, given a target speech with a particular style and ambient environment, we learn an *autoencoder specific to that target speech* (Fig. 2-c). We deem our approach "Exemplar Autoencoders". At test time, we demonstrate that one can translate any input speech into the target simply by passing it through the target exemplar autoencoder. We demonstrate this property is a consequence of two curious facts, shown in Fig. 3: (1) linguistic phonemes tend to cluster quite well in spectrogram space (Fig. 3-a); and (2) autoencoders with sufficiently small bottlenecks act as projection operators that project out-of-sample source data onto the target training distribution, allowing us to preserve the content (words) of the source and the style of the target (Fig. 3-c). Finally, we jointly synthesize audiovisual (AV) outputs by adding a visual stream to the audio autoencoder. Importantly, our approach is data-efficient and can be trained using 3 minutes of audio-video data of the target speaker and *no training data* for the input speaker. The ability to train exemplar autoencoders on small amounts of data is crucial when learning specialized models tailored to particular target data. Table 1 contrasts our work with leading approaches in audio conversion (Kaneko et al., 2019c; Qian et al., 2019) and audio-to-video synthesis (Chung et al., 2017; Suwajanakorn et al., 2017).

**Contributions:** (1) We introduce exemplar autoencoders, which allow for any input speech to be converted into an arbitrarily-large number of target speakers ("any-to-many" AV synthesis). (2) We move beyond well-curated datasets and work with in-the-wild web audio-video data in this paper. We also provide a new CelebAudio dataset for evaluation. (3) Our approach can be used as an off-the-shelf plug and play tool for target-specific voice conversion. (4) Finally, because our approach generates high-fidelity audio and video content that could be potentially misused, we discuss broader impacts in the appendix, including forensic experiments that suggests fake content can be identified with high accuracy.

## 2 RELATED WORK

A tremendous interest in audio-video generation for health-care, quality-of-life improvement, educational, and entertainment purposes has influenced a wide variety of work in audio, natural language processing, computer vision, and graphics literature. In this work, we seek to explore a standard representation for a user-controllable "any-to-many" audiovisual synthesis.

**Speech Synthesis & Voice Conversion:** Earlier works (Hunt & Black, 1996; Zen et al., 2009) in speech synthesis use text inputs to create Text-to-Speech (TTS) systems. Sequence-to-sequence (Seq2seq) structures (Sutskever et al., 2014) have led to significant advancements in TTS sys-

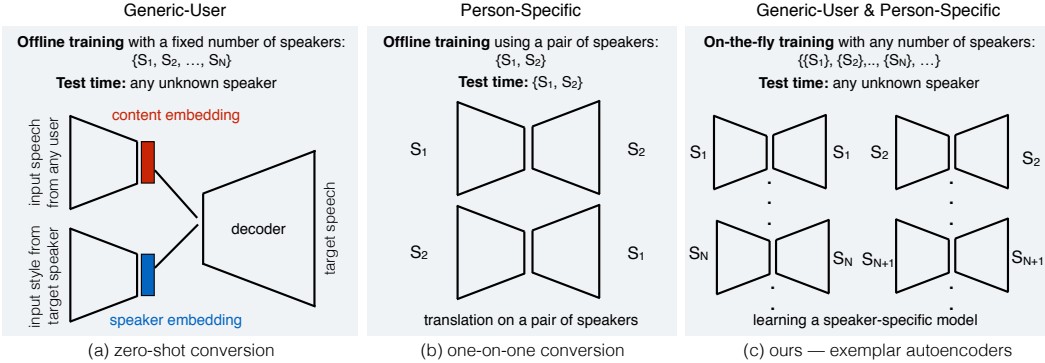

Figure 2: Prior approaches can be classified into two groups: (a) **Zero-shot conversion** learns a generic low-dimensional embedding from a training set that is designed to be agnostic to speaker identity. We empirically observe that such generic embeddings may struggle to capture stylistic details of in-the-wild speech that differs from the training set. (b) Person-specific **one-on-one conversion** learns a translation engine specialized to specific input and output speakers, restricting them to known input speakers at test time. (c) In this work, we combine the zero-shot nature of generic embeddings with the stylistic detail of person-specific translation systems. Simply put, given a target speech with a particular style and ambient environment, we learn an **autoencoder** specific to that target speech. At test time, one can translate any input speech into the target simply by passing it through the target **exemplar autoencoder**.

tems (Oord et al., 2016; Shen et al., 2018; Wang et al., 2017). Recent works (Jia et al., 2018) have extended these models to incorporate multiple speakers. These approaches enable audio conversion by generating text from the input via a speech-to-text (STT), and use a TTS for target audio. Despite enormous progress in building TTS systems, it is not trivial to embody perfect emotion and prosody due to the limited expressiveness of bare text (Pitrelli et al., 2006). In this work, we use the raw speech signal of a target speech to encode the stylistic nuance and subtlety that we wish to synthesize.

**Audio-to-Audio Conversion:** The problem of audio-to-audio conversion has largely been confined to one-to-one translation, be it using a paired (Chen et al., 2014; Nakashika et al., 2014; Sun et al., 2015; Toda et al., 2007) or unpaired (Chen et al., 2019; Kaneko & Kameoka, 2017; Kaneko et al., 2019a;b; Serrà et al., 2019; Tobing et al., 2019; Zhao et al., 2019) data setup. Recent works (Chou et al., 2019; Mohammadi & Kim, 2019; Paul et al., 2019; Polyak & Wolf, 2019; Qian et al., 2019) have begun to explore any-to-any translation, where the goal is to generalize to any input and any target speaker. To do so, such approaches learn a speaker-agnostic embedding space that is meant to generalize to never-before-seen identities. Our work is closely inspired by such approaches, but is based on the observation that such generic embeddings tend to be low-dimensional, making it difficult to capture the full range of stylistic prosody and ambient environments that can exist in speech. We can, however, capture these subtle but important aspects via an exemplar autoencoder that is trained for a specific target speech exemplar. Moreover, it is not clear how to extend such generic embeddings to audiovisual generation because visual appearance is highly multimodal (people can appear wildly different due to clothing). Exemplar autoencoders, on the other hand, are straightforward to extend to video because they inherently are tuned to the particular visual mode (clothing) in the target speech.

**Audio-Video Synthesis:** There is a growing interest (Chung et al., 2018; Nagrani et al., 2018; Oh et al., 2019) to jointly study audio and video for better recognition (Kazakos et al., 2019), localizing sound (Gao et al., 2018; Senocak et al., 2018), or learning better visual representations (Owens & Efros, 2018; Owens et al., 2016). There is a wide literature (Berthouzoz et al., 2012; Bregler et al., 1997; Fan et al., 2015; Fried et al., 2019; Greenwood et al., 2018; Taylor et al., 2017; Thies et al., 2020; Wang et al., 2021; Kumar et al., 2020) on synthesizing videos (talking-heads) from an audio signal. Early work (Yehia et al., 2002) links speech acoustics with coefficients necessary to animate natural face and head motion. Recent approaches (Chung et al., 2017; Zhou et al., 2019; Zhu et al., 2018) have looked at zero-shot facial synthesis from audio signals. These approaches register human faces using facial keypoints so that one can expect a certain facial part at a fixed location. At test time, an image of the target speaker and audio is provided. While these approaches

| Method | input: audio, **output**: ? | unknown test speaker | speaker-specific model |
|---|---|---|---|
| Auto-VC (Qian et al., 2019) | A | ✓ | ✗ |
| StarGAN-VC (Kaneko et al., 2019b) | A | ✗ | ✓ |
| Speech2Vid (Chung et al., 2017) | V | ✓ | ✗ |
| Synthesizing Obama (Suwajanakorn et al., 2017) | V | ✗ | ✓ |
| Ours (Exemplar Autoencoders) | AV | ✓ | ✓ |

Table 1: We contrast our work with leading approaches in audio conversion (Kaneko et al., 2019b; Qian et al., 2019) and audio-to-video generation (Chung et al., 2017; Suwajanakorn et al., 2017). Unlike past work, we generate both audio-video output from an audio input (**left**). Zero-shot methods are attractive because they can generalize to unknown target speakers at test time (**middle**). However, in practice, models specialized to particular pairs of known input and target speakers tend to produce more accurate results (**right**). Our method combines the best of both worlds by tuning an exemplar autoencoder on-the-fly to the target speaker of interest.

can animate lips of the target speaker, it is still difficult to capture other realistic facial expressions. Other approaches (Ginosar et al., 2019; Shlizerman et al., 2018; Suwajanakorn et al., 2017) use large amount of training data and supervision (in the form of keypoints) to learn person-specific synthesis models. While these generate realistic results, they are restricted to known input users at test time. In contrast, our goal is to synthesize audiovisual content (given *any* input audio) in a data-efficient manner (using 2-3 minutes of unsupervised data).

**Autoencoders:** Autoencoders are unsupervised neural networks that learn to reconstruct inputs via a bottleneck layer (Bengio et al., 2017). They have traditionally been used for dimensionality reduction or feature learning, though variational formulations have extended them to full generative models that can be used for probabilistic synthesis (Kingma & Welling, 2013). Most related to us are denoising autoencoders (Vincent et al., 2008), which learn to project noisy input data onto the underlying training data manifold. Such methods are typically trained on synthetically-corrupted inputs with the goal of learning useful representations. Instead, we exploit the projection property of autoencoders itself, by repurposing them as translation engines that "project" the input speech of an unknown speaker onto the target speaker manifold. It is well-known that linear autoencoders, which can be learned with PCA (Baldi & Hornik, 1989), produce the best reconstruction of any out-of-sample input datapoint, in terms of squared error from the subspace spanned by the training dataset (Bishop, 2006). We empirically verify that this property approximately holds for nonlinear autoencoders (Fig. 3). When trained on a target (exemplar) dataset consisting of a particular speech style, we demonstrate that reprojections of out-of-sample inputs tend to preserve the content of the input and the style of the target.

## 3 EXEMPLAR AUTOENCODERS

There is an enormous space of audiovisual styles spanning visual appearance, prosody, pitch, emotions, and environment. It is challenging for a single *large* model to capture such diversity for accurate audiovisual synthesis. However, many *small* models may easily capture the various nuances. In this work, we learn a separate autoencoder for individual audiovisual clips that are limited to a particular style. We call our technique Exemplar Autoencoders. One of our remarkable findings are that, when trained appropriately, Exemplar Autoencoders can be driven with the speech input of a never-before-seen speaker, allowing for the overall system to function as a audiovisual translation engine. We study in detail why audiovisual autoencoders preserve the *content* (words) of never-before-seen input speech but preserve the *style* of the target speech on which it was trained. The heart of our our approach relies on the compressibility of audio speech, which we describe below.

Speech contains two types of information: (i) the **content**, or words being said and (ii) **style** information that describes the scene context, person-specific characteristics, and prosody of the speech delivery. It is natural to assume that speech is generated by the following process. First, a style $s$ is drawn from the style space $S$. Then a content code $w$ is drawn from the content space $W$. Fi-

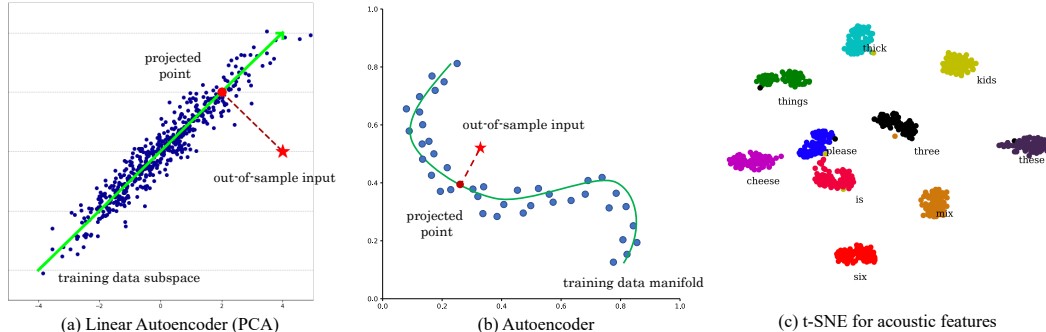

(a) Linear Autoencoder (PCA)  (b) Autoencoder  (c) t-SNE for acoustic features

Figure 3: **Autoencoders** with sufficiently small bottlenecks project out-of-sample data onto the manifold spanned by the training set (Bengio et al., 2017), which can be easily seen in the case of **linear autoencoders** learned via PCA (Bishop, 2006). We observe that **acoustic features** (MEL spectograms) of words spoken by different speakers (from the VCTK dataset (Veaux et al., 2016)) easily cluster together irrespective of who said them and with what style. We combine these observations to design a simple style transfer engine that works by training an autoencoder on a *single target style* with a *MEL-spectogram reconstruction loss*. Such *exemplar autoencoders* project out-of-sample input speech content onto the style-specific manifold of the target.

nally, $x = f(s, w)$ denotes the speech of the content $w$ spoken in style $s$, where $f$ is the generating function of speech.

## 3.1 STRUCTURED SPEECH SPACE

In human acoustics, one uses different shapes of their vocal tract[2] to pronounce different words with their voice. Interestingly, different people use similar, or ideally the same, shapes of vocal tract to pronounce the same words. For this reason, in the speech space we can find a built-in structure that the acoustic features of the same words in different styles are very close (also shown in Fig. 3-a). Given two styles $s_1$ and $s_2$ for example, $f(s_1, w_0)$ is one spoken word in style $s_1$. Then $f(s_2, w_0)$ should be closer to $f(s_1, w_0)$ than any other word in style $s_2$. We can formulate this property as follows:

$$\text{Error}(f(s_1, w_0), f(s_2, w_0)) \leq \text{Error}(f(s_1, w_0), f(s_2, w)), \forall w \in W, \quad \text{where} \quad s_1, s_2 \in S.$$

This can be further presented in equation form:

$$f(s_2, w_0) = \arg \min_{x \in M} \text{Error}(x, f(s_1, w_0)), \quad \text{where} \quad M = \{f(s_2, w) : w \in W\}. \quad (1)$$

## 3.2 AUTOENCODERS FOR STYLE TRANSFER

We now provide a statistical motivation for the ability of exemplar autoencoders to preserve content while transferring style. Given training examples $\{x_i\}$, one learns an encoder $E$ and decoder $D$ so as to minimize reconstruction error:

$$\min_{E,D} \sum_i \text{Error}\Big(x_i, D(E(x_i))\Big).$$

In the linear case (where $E(x) = Ax$, $D(x) = Bx$, and Error = L2), optimal weights are given by the eigenvectors that span the input subspace of data (Baldi & Hornik, 1989). Given sufficiently small bottlenecks, linear autoencoders project out-of-sample points into the input subspace, so as to minimize the reconstruction error of the output (see Fig. 3-(a)). Weights of the autoencoder (eigenvectors) capture dataset-specific *style* common to all samples from that dataset, while the bottleneck activations (projection coefficients) capture sample-specific *content* (properties that capture individual differences between samples). In the appendix, we empirically verify that a similar property

---

[2]The vocal tract is the cavity in human beings where the sound produced at the sound source is filtered. The shape of the vocal tract is mainly determined by the positions and shapes of the tongue, throat and mouth.

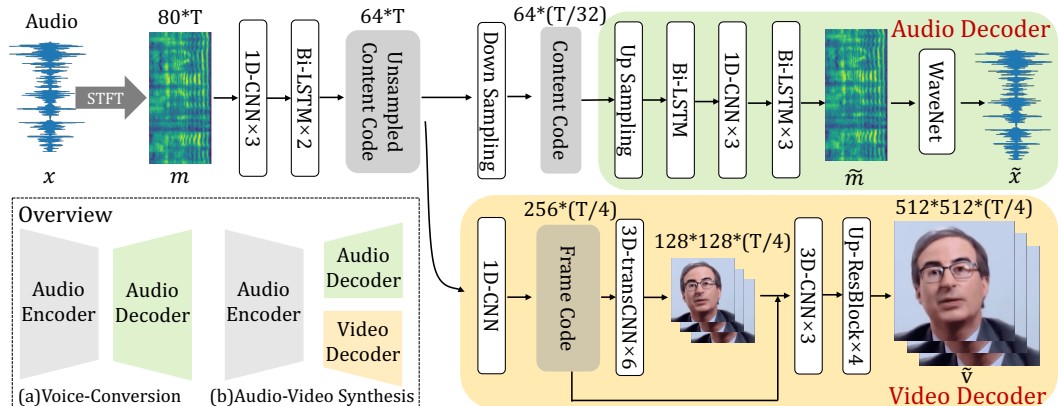

Figure 4: **Network Architecture:** (a) The voice-conversion network consists of a content encoder, and an audio decoder (denoted as green). This network serves as a person & attributes-specific auto-encoder at training time, but is able to convert speech from anyone to personalized audio for the target speaker at inference time. (b) The audio-video synthesis network incorporates a video decoder (denoted as yellow) into the voice-conversion system. The video decoder also regards the content encoder as its front-end, but takes the unsampled content code as input due to time alignment. The video architecture is mainly borrowed from StackGAN (Radford et al., 2016; Zhang et al., 2017), which synthesizes the video through 2 resolution-based stages.

holds for nonlinear autoencoders: given sufficiently-small bottlenecks, they approximately project out-of-sample data onto the nonlinear *manifold* $M$ spanned by the training set:

$$D(E(\hat{x})) \approx \arg \min_{m \in M} \text{Error}(m, \hat{x}), \text{where} \qquad M = \{D(E(x)) | x \in \mathbb{R}^d\}. \qquad (2)$$

where the approximation is exact for the linear case. In the nonlinear case, let $M = \{f(s_2, w) : w \in W\}$ be the manifold spanning a particular style $s_2$. Let $\hat{x} = f(s_1, w_0)$ be a particular word $w_0$ spoken with any other style $s_1$. The output $D(E(\hat{x}))$ will be a point on the target manifold $M$ (roughly) closest to $\hat{x}$. From Eq. 1 and Fig. 3, the closest point on the target manifold tends to be the same word spoken by the target style:

$$D(E(\hat{x})) \approx \arg \min_{t \in M} \text{Error}(t, \hat{x}) = \arg \min_{t \in M} \text{Error}(t, f(s_1, w_0)) \approx f(s_2, w_0). \qquad (3)$$

Note that nothing in our analysis is language-specific, so it still holds, in principle, for other languages such as Mandarin and Hindi. We posit that the "content" $w_i$ can be represented by phonemes that can be shared by different languages. In our summary video, we verify this by showing that audiovisual translation is possible *across* different languages; one can drive John Oliver to speak in Chinese or Hindi.

## 3.3 STYLISTIC AUDIOVISUAL (AV) SYNTHESIS

We now operationalize our previous analysis into an approach for stylistic AV synthesis. To do so, we learn AV representations with autoencoders tailored to particular target speech. To enable these representations to be driven by the audio input of any user, we learn in a manner that ensures bottleneck activations capture structured linguistic "word" content: we pre-train an audio-only autoencoder, and then learn a video decoder while finetuning the audio component. This adaptation allows us to train AV models for a target identity with as little as 3 minutes of data (in contrast to the 14 hours used in (Suwajanakorn et al., 2017)).

**Audio autoencoder:** Given the target audio stream of interest $x$, we first convert it to a mel spectogram $m = \mathcal{F}(x)$ using a short-time Fourier Transform (Oppenheim et al., 1996). We train an encoder $E$ and decoder $D$ that reconstructs Mel-spectograms $\tilde{m}$, and finally use a WaveNet vocoder $V$ (Oord et al., 2016) to convert $\tilde{m}$ back to speech $\tilde{x}$. Importantly, we train all components (the encoder $E$, decoder $D$, and vocoder $V$) with a joint reconstruction loss in both frequency and audio space:

$$\text{Error}_{Audio}(x, \tilde{x}) = \mathbb{E} \| m - \tilde{m} \|_1 + L_{WaveNet}(x, \tilde{x}), \qquad (4)$$

| VCTK (Veaux et al., 2016) | Zero-Shot | Extra-Data | SCA (%) ↑ | MCD ↓ |
|---|---|---|---|---|
| StarGAN-VC (Kaneko et al., 2019b) | ✗ | ✓ | 69.5 | 582.1 |
| VQ-VAE (van den Oord et al., 2017) | ✗ | ✓ | 69.9 | 663.4 |
| Chou et al. (Chou et al., 2018) | ✗ | ✓ | 98.9 | **406.2** |
| Blow (Serrà et al., 2019) | ✗ | ✓ | 87.4 | 444.3 |
| Chou et al. (Chou et al., 2019) | ✓ | ✓ | 57.0 | 491.1 |
| Auto-VC (Qian et al., 2019) | ✓ | ✓ | 98.5 | 408.8 |
| Ours | ✓ | ✗ | **99.6** | 420.3 |

Table 2: **Objective Evaluation for Audio Translation:** We evaluate on VCTK, which provides paired data. The speaker-classification accuracy (SCA) criterion enables us to study the naturalness of generated audio samples and similarity to the target speaker, where **higher is better**. Typical methods measure reoconstruction error with Mel-Cepstral distortion (MCD) (where **lower is better**), but our prior analysis show that this will be dominated by matching the content words (Fig. 3). Our approach achieves competitive performance to prior state-of-the-art without requiring additional data from other speakers and yet be zero-shot. We do a more comprehensive human study in Table 3 using CelebAudio dataset to study the influence of data.

where $L_{WaveNet}$ is the standard cross-entropy loss used to train Wavenet (Oord et al., 2016). Fig. 4 (top-row) summarizes our network design for audio-only autoencoders. Additional training parameters are provided in the appendix.

**Video decoder:** Given the trained audio autoencoder $E(x)$ and a target video $v$, we now train a video decoder that reconstructs the video $\tilde{v}$ from $E(x)$ (see Fig. 4). Specifically, we train using a joint loss:

$$\text{Error}_{AV}(x, v, \tilde{x}, \tilde{v}) = \text{Error}_{Audio}(x, \tilde{x}) + \mathbb{E}\|v - \tilde{v}\|_1 + L_{Adv}(v, \tilde{v}). \tag{5}$$

where $L_{Adv}$ is an adversarial loss (Goodfellow et al., 2014) used to improve video quality. We found it helpful to simultaneously fine-tune the audio autoencoder and so add equation 4 to the overall AV loss.

## 4 EXPERIMENTS

We now quantitatively evaluate the proposed method for audio conversion in Sec. 4.1 and audio-video synthesis in Sec. 4.2. We attempted to find all public codebases that could serve as baselines for audio conversion and audiovisual generation. We use existing datasets for a fair evaluation with prior art. We also introduce a new challenging dataset that consists of recordings in non-studio environments. Many of our motivating applications, such as education and assistive technology, require processing unstructured real-world data collected outside a studio. We urge the reader to see our summary video to see/hear our results, as well as the appendix for additional ablative analysis.

### 4.1 AUDIO TRANSLATION

**Datasets:** We use the publicly available **VCTK** dataset (Veaux et al., 2016), which contains 44 hours of utterances from 109 native speakers of English with various accents. Each speaker reads a different set of sentences, except for two paragraphs. While the conversion setting is unpaired, there exists a small amount of paired data that enables us to conduct objective evaluation. We also introduce the new *in-the-wild* **CelebAudio** dataset for audio translation to validate the effectiveness as well as the robustness of various approaches. This dataset consists of speeches (with an average of 30 minutes, but some as low as 3 minutes) of various public figures collected from YouTube, including native and non-native speakers. The content of these speeches is entirely different from one another, thereby forcing the future methods to be unpaired and unsupervised. There are 20 different identities. We provide more details about CelebAudio dataset in the appendix.

**Quantitative Evaluation:** We rigorously evaluate our output with a variety of metrics. When ground-truth paired examples are available for testing (VCTK), we follow past work (Kaneko et al., 2019a;b) and compute Mel-Cepstral distortion (**MCD**), which measures the squared distance between the synthesized audio and ground-truth in frequency space. Inspired by "FCN-score" metrics for image generation, we propose to use speaker-classification accuracy (**SCA**), which is defined as

| CelebAudio | Extra Data | Naturalness ↑ | | Voice Similarity ↑ | | Content Consistency ↑ | | Geometric Mean of VS-CC ↑ |
|---|---|---|---|---|---|---|---|---|
| | | MOS | Pref (%) | MOS | Pref (%) | MOS | Pref (%) | |
| **Auto-VC** (Qian et al., 2019) | | | | | | | | |
| off-the-shelf | - | 1.21 | 0 | 1.31 | 0 | 1.60 | 0 | 1.45 |
| fine-tuned | ✓ | 2.35 | 13.3 | 1.97 | 2.0 | **4.28** | **48.0** | 2.90 |
| scratch | ✗ | 2.28 | 6.7 | 1.90 | 2.0 | 4.05 | 18.0 | 2.77 |
| **Ours** | ✗ | **2.78** | **66.7** | **3.32** | **94.0** | 4.00 | 22.0 | **3.64** |

Table 3: **Human Studies for Audio**: We conduct extensive human studies on Amazon Mechanical Turk. We report Mean Opinion Score (MOS) and user preference (percentage of time that method ranked best) for naturalness, target voice similarity (VS), source content consistency (CC), and the geometric mean of VS-CC (since either can be trivially maximized by reporting a target/input sample). Higher the better, on a scale of 1-5. The off-the-shelf Auto-VC model struggles to generalize to CelebAudio, indicating the difficulty of in-the-wild zero-shot conversion. Fine-tuning on CelebAudio dataset significantly improves performance. When restricting Auto-VC to the same training data as our model (scratch), performance drops a small but noticeable amount. Our results strongly outperform Auto-VC for VS-CC, suggesting exemplar autoencoders are able to generate speech that maintains source content consistency while being similar to the target voice style. We provide more ablation analysis in the Appendix.

the percentage of times a translation is correctly classified by a speaker-classifier (trained with Serrà et al. (2019)).

**Human Study Evaluation:** We conduct extensive human studies on Amazon Mechanical Turk (AMT) to analyze the generated audio samples from CelebAudio. The study assesses the naturalness of the generated data, the voice similarity (VS) to the target speaker, the content/word consistency (CC) to the input, and the geometric mean of VS-CC (since either can be trivially maximized by reporting a target/input speech sample). Each is measured on a scale from 1-5. We select 10 CelebAudio speakers as targets and randomly choose 5 utterances from the other speakers as inputs. We then produce $5 \times 10 = 50$ translations, each of which is evaluated by 10 AMT users.

**Baselines:** We contrast our method with several existing voice conversion systems (Chou et al., 2018; Kaneko et al., 2019b; van den Oord et al., 2017; Qian et al., 2019; Serrà et al., 2019). Because StarGAN-VC (Kaneko et al., 2019b), VQ-VAE (van den Oord et al., 2017), Chou et al. (Chou et al., 2018), and Blow (Serrà et al., 2019) do not claim zero-shot voice conversion, we train these models on 20 speakers from VCTK and evaluate voice conversion on those speakers. Table 2 shows that we outperform these approaches for speaker classification, while being competitive for MCD reconstruction error.

**Auto-VC (Qian et al., 2019):** Auto-VC is a zero-shot translator. It performs quite well in SCA and MCD metrics in Table 2, and so we extensively explore it via AMT human studies in Table 3. Since Auto-VC claims any-to-any audio conversion, we first use an **off-the-shelf** model for evaluation. We observe poor performance, both quantitatively and qualitatively. We then **fine-tune** the existing model on audio data from 20 CelebAudio speakers. We observe significant performance improvement in Auto-VC when restricting it to the same set of examples as ours. We also trained the Auto-VC model from **scratch**, for an apples-to-apples comparison with us. The performance on all three criterion dropped with lesser data. On the other hand, our approach generates significantly better audio which sounds more like the target speaker while still preserving the content.

## 4.2 AUDIO-VIDEO SYNTHESIS

**Dataset:** In addition to augmenting CelebAudio with video clips, we also evaluate on VoxCeleb (Chung et al., 2018), an audio-visual dataset of short celebrity interview clips from YouTube.

**Baselines:** We compare to **Speech2Vid** (Chung et al., 2017), using their publicly available code. This approach requires the face region to be registered before feeding it to a pre-trained model for facial synthesis. We find various failure cases where the human face from in-the-wild videos of VoxCeleb dataset cannot be correctly registered. Additionally, Speech2Vid generates the video of a cropped face region by default. We composite the output on a still background to make it more compelling and comparable to ours. Finally, we also compare to the recent work of **LipGAN** (KR

et al., 2019). This approach synthesizes region around lips and paste this generated face crop into the given video. We observe that the paste is not seamless and leads to artifacts especially when working with in-the-wild web videos. We show comparisons with both Speech2Vid (Chung et al., 2017) and LipGAN (KR et al., 2019) in Figure 6. The artifacts from both approaches are visible when used for in-the-wild video examples from CelebAudio and VoxCeleb dataset. However, our approach generates the complete face without these artifacts and captures articulated expressions.

**Human Studies:** We conducted an AB-Test on AMT: we show a pair of our video output and another method to the user, and ask the user to select which looks better. Each pair is shown to 5 users. Our approach is preferred 87.2% over Speech2Vid (Chung et al., 2017) and 59.3% over LipGAN (KR et al., 2019).

## 5 DISCUSSION

In this paper, we propose exemplar autoencoders for unsupervised audiovisual synthesis from speech input. Our model is able to generalize to in-the-wild web data and diverse forms of speech input, enabling novel applications in entertainment, education, and assistive technology.

Our work opens up a rethinking of autoencoders' modeling capacity. Under certain conditions, autoencoders serve as projective operators to some specific distribution.

In the appendix, we provide an extensive discussion of broader impacts including potential abuses of such technology. We explore mitigation strategies, including empirical evidence that forensic classifiers can be used to detect synthesized content from our method.

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

# A  APPENDIX - BROADER IMPACTS

Our work falls in line with a body of work on content generation that retargets video content, often considered in the context of facial puppeteering. While there exist many applications in entertainment, there also exist many potentials for serious abuse. Past work in this space has included broader impact statements (Fried et al., 2019; Kim et al., 2018), which we build upon here.

**Our Setup:** Compared to prior work, unique aspects of our setup are (a) the choice of input and output modalities and (b) requirements on training data. In terms of (a), our models take audio as input, and produce audiovisual (AV) output that captures the personalized style of the target individual but the linguistic content (words) of the source input. This factorization of style and content across the source and target audio is a key technical aspect of our work. While past work has discussed broader impacts of visual image/video generation, there is less discussion on responsible practices for audio editing. We begin this discussion below. In terms of (b), our approach is unique in that we require only a few minutes of training on target audio+video, and do not require training on large populations of individuals. This has implications for generalizability and ease-of-use for non-experts, increasing the space of viable target applications. Our target applications include entertainment/education and assistive technologies, each of which is discussed below. We conclude with a discussion of potential abuses and strategies for mitigation.

**Assistive Technology:** An important application of voice synthesis is voice generation for the speaking impaired. Here, generating speech in an individuals personalized style can be profoundly important for maintaining a sense of identity[3]. We demonstrate applications in this direction in our summary videos. We have also begun collaborations with clinicians to explore personalized and stylistic speech outputs from physical devices (such as an electrolarynx) that directly sense vocal utterances. Currently, there are considerable data and privacy considerations in acquiring such sensitive patient data. We believe that illustrating results on well-known individuals (such as celebrities) can be used to build trust in clinical collaborators and potential patient volunteers.

**Entertainment/Education:** Creating high-quality AV content is a labor intensive task, often requiring days/weeks to create minutes of content for production houses. Our work has attracted the attention of production houses that are interested in creating documentaries or narrations of events by historical figures in their own voice (Winston Churchill, J F Kennedy, Nelson Mandela, Martin Luther King Jr). Because our approach can be trained on small amounts of target footage, it can be used for personalized storytelling (itself a potentially creative and therapeutic endeavour) as well as educational settings that have access to less computational/artistic resources but more immediate classroom feedback/retraining.

**Abuse:** It is crucial to acknowledge that audiovisual retargeting can be used maliciously, including spreading false information for propaganda purposes or pornographic content production. Addressing these abuses requires not only technical solutions, but discussions with social scientists, policy makers, and ethicists to help delineate boundaries of defamation, privacy, copyright, and fair use. We attempt to begin this important discussion here. First, inspired by (Fried et al., 2019), we introduce a recommended policy for AV content creation using our method. In our setting, retargeting involves two pieces of media; the source individual providing an audio signal input, and target individual whos audiovisual clip will be edited to match the words of the source. Potentials for abuse include plagiarism of the source (e.g., representing someone elses words as your own), and misrepresentation / copyright violations of the target. Hence, we advocate a policy of always citing the source content and always obtaining permission from the target individual. Possible exceptions to the policy, if any, may fall into the category of fair use[4], which typically includes education, commentary, or parody (eg., retargeting a zebra texture on Putin (Zhu et al., 2017)). In all cases, retargeted content should acknowledge itself as edited, either by clearly presenting itself as parody or via a watermark. Importantly, there may be violations of this policy. Hence, a major challenge will be identifying such violations and mitigating the harms caused by such violations, discussed further below.

---

[3]https://news.northeastern.edu/2019/11/18/personalized-text-to-speech-voices-help-people-with-speech-disabilities-maintain-identity-and-social-connection/

[4]https://fairuse.stanford.edu/overview/fair-use/what-is-fair-use/

**Audiovisual Forensics:** We describe 3 strategies that attempt to identify misuse and the harms caused by such misuse. We stress that these are not exhaustive: (a) identifying "fake" content, (b) data anonymization, and (c) technology awareness. (a) As approaches for editing AV content mature, society needs analogous approaches for detecting such "fake" content. Contemporary forensic detectors tend to be data-driven, themselves trained to distinguish original versus synthesized media via a classification task. Such forensic real-vs-fake classifiers exist for both images (Maximov et al., 2020; Rössler et al., 2019; Wang et al., 2020) and audio[5,6]. Because access to code for generating "fake training examples will be crucial for learning to identify fake content, we commit to making our code freely available. Appendix B provides extensive analysis of audio detection of Exemplar Autoencoder fakes. (b) Another strategy for mitigating abuse is controlling access to private media data. Methods for image anonymization, via face detection and blurring, are widespread and crucial part of contemporary data collection, even mandated via the EUs General Data Protection Regulation (GDPR). In the US, audio restrictions are even more severe because of federal wiretapping regulations that prevent recordings of oral communications without prior consent (Stevens & Doyle, 2011). Recent approaches for image anonymization make use of generative models that de-identify data without degradive blurring by retargeting each face to a generic identity (e.g., make everyone in a dataset look like PersonA) (Maximov et al., 2020). Our audio retargeting approach can potentially be used for audio de-identification by making everyone in a recording **sound** like PersonA. (c) Finally, we point out that audio recordings are currently used in critical societal settings including legal evidence and biometric voice-recognition (e.g., accessing ones bank account via automated recognition of speech over-the-phone (Stevens & Doyle, 2011)). Our results suggest that such use cases need to be re-evaluated and thoroughly tested in context of stylistic audio synthesis. In other terms, it is crucial for society to understand what information can be reliably deduced from audio, and our approach can be used to empirically explore this question.

**Training datasets:** Finally, we point out a unique property of our technical approach that differs prior approaches for audio and video content generation. Our exemplar approach reduces the reliance on population-scale training datasets. Facial synthesis models trained on existing large-scale datasets – that may be dominated by english-speaking celebrities – may produce higher accuracy on sub-populations with skin tones, genders, and languages over-represented in the training dataset (Menon et al., 2020). Exemplar-based learning may exhibit different properties because models are trained on the target exemplar individual. That said, we do find that pre-training on a single individual (but not a population) can speed up convergence of learning on the target individual. Because of the reduced dependency on population-scale training datasets (that may be dominated by English), exemplar models may better generalize across dialects and languages underrepresented in such datasets. In our summary video, we present results for stylized multilingual translations (retargeting Oliver to speak in Mandarin and Hindi) without ever training on any Mandarin or Hindi speech.

## B  APPENDIX - FORENSIC STUDY

In the previous section, we outline both positive and negative outcomes of our research. Compared to prior art, most of the novel outcomes arise from using audio as an additional modality. As such, in this section, we conduct a study which illustrates that Exemplar Autoencoder fakes can be detected with high accuracy by a forensic classifier, particularly when trained on such fakes. We hope our study spurs additional forensic research on detection of manipulated audiovisual content.

**Speaker Agnostic Discriminator:** We begin by training a real/fake audio classifier on 6 identities from CelebAudio. The model structure is the same as (Serrà et al., 2019). The training datasets contains (1) Real speech of 6 identities: John F Kennedy, Alexei Efros, Nelson Mandela, Oprah Winfrey, Bill Clinton, and Takeo Kanade. Each has 200 sentences. (2) Fake speech: generated by turning each sentence in (1) into a different speaker in those 6 identities. So there is a total of 1200 generated sentences. We test performance under 2 scenarios: (1) speaker within-the-training set: John F Kennedy, Alexei Efros, Nelson Mandela, Oprah Winfrey, Bill Clinton, and Takeo Kanade; (2) speaker-out-of-the-training set: Barack Obama, Theresa May. Each identity in the test set contains

---

[5]https://www.blog.google/outreach-initiatives/google-news-initiative/advancing-research-fake-audio-detection/

[6]https://www.asvspoof.org/

| Speaker-Agnostic Discriminator | Accuracy(%) | Speaker-Specific Discriminator | Accuracy(%) |
|---|---|---|---|
| **speaker within training set** | 99.0 | **style within training set** | 99.7 |
| real | 98.3 | real | 99.3 |
| fake | 99.7 | fake | 100 |
| **speaker out of training set** | 86.8 | **style out of training set** | 99.6 |
| real | 99.7 | real | 99.2 |
| fake | 74.0 | fake | 100 |

Table 4: **Fake Audio Discriminator:** a) **Speaker Agnostic Discriminator** detects fake audio while agnostic of the speaker. We train a real/fake audio classifier on 6 identities from CelebAudio, and test it on both within-set speakers and out-of-set speakers. The training datasets contains (1) Real speech of 6 identities: John F Kennedy, Alexei Efros, Nelson Mandela, Oprah Winfrey, Bill Clinton, and Takeo Kanade. Each has 200 sentences. (2) Fake speech: generated by turning each sentence in (1) into a different speaker in those 6 identities. The testing set contains (1) speaker within the training set; (2) speaker out of the training set: Barack Obama, Theresa May. Each identity in the testing set contains 100 real sentences and 100 fake ones. For within-set speakers, results show our model can predict with very low error rate ( 1%). For out-of-set speakers, our model can still classify real speeches very well, and provide a reasonable prediction for fake detection. b) **Speaker Specific Discriminator** detects fake audio of a specific speaker. We train a real/fake audio classifier on a specific style of Barack Obama, and test it on 4 styles of Obama (taken from speeches spanning different ambient environments and stylistic deliveries including presidential press conferences and university commencement speechs). The training set contains 1200 sentences evenly split between real and fake. Each style in the testing set contains 300 sentences evenly split as well. Results show our classifier provides reliable predictions on fake speeches even on out-of-sample styles.

100 real sentences and 100 fake ones. We ensure test speeches are disjoint from training speeches, even for the same speaker. Table 4 shows the classifier performs very well on detecting fake of within-set speakers, and is able to provide a reasonable reference for out-of-sample speakers.

**Speaker-Specific Discriminator:** We restrict our training set to one identity for specific fake detection. We train 4 exemplar autoencoders on 4 different speeches of Barack Obama to get different styles of the same person. The specific fake audio detector is trained on only 1 style of Obama, and is tested on all the 4 styles. The training set contains 600 sentences for either real or fake. Each style in the testing set contains 150 sentences for either real or fake. Table 4 shows our classifier provides reliable predictions on fake speeches even on out-of-set styles.

## C  APPENDIX - IMPLEMENTATION DETAILS

We illustrate our method in Fig. 4 in main paper. Below we provide the implementation details of our exemplar autoencoder.

### C.1  AUDIO CONVERSION

**STFT:** The speech data is sampled at 16 kHz. We clip the training speech into clips of 1.6s in length, which correspond to 25,600-dimensional vectors. We then perform STFT(Oppenheim et al., 1996) on the raw audio signal with a window size of 800, hop size of 200, and 80 mel-channels. The output of STFT is a complex matrix with size of $80 \times 128$. We represent the complex matrix in polar form (magnitude and phase), and only keep the magnitude for next steps.

**Encoder:** The input to the encoder is the magnitude of $80 \times 128$ Mel-spectrogram, which is represented as a 1D 80-channel signal (shown in Figure 4 in main paper). This input is feed-forward to three layers of 1D convolutional layers with a kernel size of 5, each followed by batch normalization (Ioffe & Szegedy, 2015) and ReLU activation (Krizhevsky et al., 2012). The channel of these convolutions is 512. The stride is one. There is no time down-sampling up till this step. The output is then fed into two layers of bidirectional LSTM (Hochreiter & Schmidhuber, 1997) layers with both the forward and backward cell dimensions of 32. We then perform a different down-sampling for the forward and backward paths with a factor of 32 following (Qian et al., 2019). The result content embedding is a matrix with a size of $64 \times 4$.

**Audio Decoder:**  The content embedding is up-sampled to the original time resolution of $T$. The up-sampled embedding is sequentially input to a 512-channel LSTM layer and three layers of 512-channel 1D convolutional layers with a kernel size of 5. Each step accompanies batch normalization and ReLU activation. Finally, the output is fed into two 1024-channel LSTM layers and a fully connected layer to project into 80 channels. The projection output is regarded as the generated magnitude of Mel-spectrogram $\tilde{m}$.

**Vocoder:**  We use WaveNet as vocoder (Oord et al., 2016) that acts like an inverse fourier-transform, but merely use frequency magnitudes. It generates speech signal $\tilde{x}$ based on the reconstructed magnitude of Mel-spectrogram $\tilde{m}$.

**Training Details:**  Our model is trained at a learning rate of 0.001 and a batch size of 8. To train a model from scratch, it needs about 30 minutes of the target speaker's speech data and around 10k iterations to converge. Although our main structure is straightforward, the vocoder is usually a large and complicated network, which needs another 50k iterations to train. However, transfer learning can be beneficial in reducing the number of iterations and necessary data for training purposes. When fine-tuning a new speaker's autoencoder from a pre-trained model, we only need about 3 minutes of speech from a new speaker. The entire model, including the vocoder, converges around 10k iterations.

## C.2 VIDEO SYNTHESIS

**Network Architecture:**  We keep the voice-conversion framework unchanged and enhance it with an additional audio-to-video decoder. In the voice-conversion network, we have a content encoder that extracts content embedding from speech, and an audio decoder that generates audio output from that embedding. To include video synthesis, we add a video decoder which also takes the content embedding as input, but generates video output instead. As shown in Figure 4 (bottom-row) in main paper, we then have an audio-to-audio-video pipeline.

**Video Decoder:**  We borrow architecture of the video decoder from (Radford et al., 2016; Zhang et al., 2017). We adapt this image synthesis network by generating the video frame by frame, as well as replacing the 2D-convolutions with 3D-convolutions to enhance temporal coherence. The video decoder takes the unsampled content codes as input. This step is used to align the time resolution with 20-fps videos in our experiments. We down-sample it with a 1D convolutional layer. This step helps smooth the adjacent video frames. The output is then fed into the synthesis network to get the video result $\tilde{v}$.

**Training Details:**  We train an audiovisual model based on a pretrained audio model, with a learning rate of 0.001 and a batch size of 8. When fine-tuning from a pre-trained model, the overall model converges around 3k iterations.

# D APPENDIX - ADDITIONAL EXPERIMENTS

## D.1 CELEBAUDIO DATASET

In Sec 4.1 of main paper, we introduce a new in-the-wild CelebAudio dataset for audio translation. Different from well-curated speech corpus such as VCTK, CelebAudio is collected from the Internet, and thus has such slight noises as applause and cheers in it. This is helpful to validate the effectiveness and robustness of various voice-conversion methods. The distribution of speech length is shown in Fig. 5. We select a subset of 20 identities for evaluation in our work. The list of 20 identities is:

- **Native English Speakers:** Alan Kay, Carl Sagan, Claude Shannon, John Oliver, Oprah Winfrey, Richard Hamming, Robert Tiger, David Attenborough, Theresa May, Bill Clinton, John F Kennedy, Barack Obama, Michelle Obama, Neil Degrasse Tyson, Margaret Thatcher, Nelson Mandela, Richard Feyman

- **Non-Native English Speakers:** Alexei Efros, Takeo Kanade

- **Computer-generated Voice:** Stephen Hawking

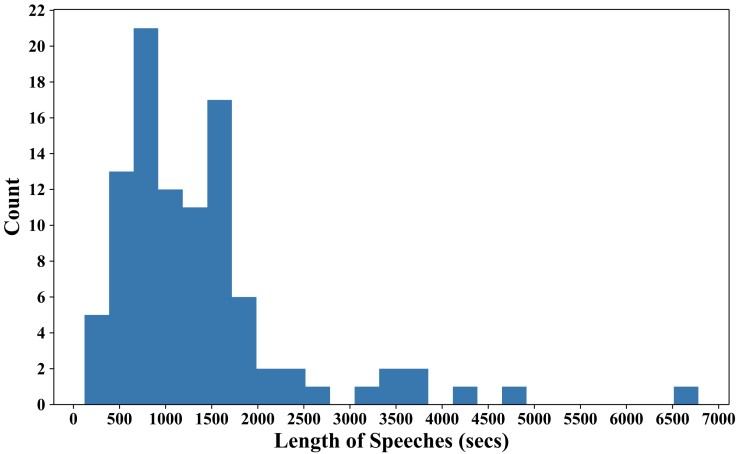

Figure 5: **Data Distribution of CelebAudio:** CelebAudio contains 100 different speeches. The average speech length is 23 mins. Each speech ranges from 3 mins (Hillary Clinton) to 113 mins (Neil Degrasse Tyson).

### D.2 VERIFICATION OF REPROJECTION PROPERTY

In Sec 3.2 of main paper, we describe a reprojection property (Eq. 6) of our exemplar autoencoder: the output of out-of-sample data $\hat{x}$ is the projection to the manifold $M$ spanned by the training set.

$$D(E(\hat{x})) \approx \arg \min_{m \in M} \text{Error}(s, \hat{x}), \text{where} \qquad M = \{D(E(x))|x \in \mathbb{R}^d\}. \tag{6}$$

This is equivalent to two important properties: (a) The output lies in the manifold $M$. (Eq. 7) (b) The output is closer to the input $\hat{x}$ than any other point on manifold $M$. (Eq. 8)

$$D(E(\hat{x})) \in M, \text{where} \qquad M = \{D(E(x))|x \in \mathbb{R}^d\}. \tag{7}$$

$$\text{Error}(D(E(\hat{x})), \hat{x}) \leq \text{Error}(x, \hat{x}), \qquad \forall x \in M. \tag{8}$$

To verify these, we conduct an experiment (Table 5) on the parallel corpus of VCTK(Veaux et al., 2016). We randomly sample 2 sets of words (normal and tongue-twister) spoken by 2 speakers (A and B) from VCTK dataset. We train an exemplar autoencoder on A's speech, and input B's words ($w_B$) to get $w_{B \rightarrow A}$. To verify Eq. 7, we train a speaker-classifier (as Serra et al. (Serrà et al., 2019)) on A and B's speech. As the fifth column in Table 5, we report the likelihood that $w_{B \rightarrow A}$ is regarded as A's words. To verify Eq. 8, we calculate $\text{Error}(D(E(\hat{x})), \hat{x})$ as the second column in Table 5, and $\text{Error}(x, \hat{x})$ as the third and fourth columns in Table 5. The results show that our conjecture about the reprojection property is reasonable.

### D.3 ABLATION ANALYSIS OF EXEMPLAR TRAINING

In the main paper, we posit that it is easier to disentangle style from content when the entire training dataset consists of a single style. To verify this, we train an autoencoder with following setup: (1) speeches from 2 people; (2) speeches from 3 people; (3) 4 speeches with different settings from 1 person. We contrast it with exemplar autoencoder trained using 1 speech from 1 person using A/B testing. Our exemplar results are preferred 71.9% times over (1), 83.3% times over (2), and 90.6% times over (3).

### D.4 ABLATION ANALYSIS OF REMOVING SPEAKER EMBEDDING

We point out that generic speaker embeddings struggles to capture stylistic details of in-the-wild speech. Here we provide the ablation analysis of removing such pre-trained embeddings.

(a) Normal words

| $w_B$ | $\|w_{B\to A} - w_B\|$ | $\min\limits_{w\in s_A}\|w - w_B\|$ | $\arg\min\limits_{w\in s_A}\|w - w_B\|$ | Likelihood(%) |
|---|---|---|---|---|
| "many" | **18.27** | 28.33 | "many" | 100 |
| "complicate" | **16.15** | 20.05 | "complicate" | 59.4 |
| "idea" | **18.54** | 26.57 | "idea" | 100 |
| "about" | **14.20** | 26.51 | "about" | 100 |
| "rainbow" | **15.68** | 27.63 | "rainbow" | 100 |
| "form" | **13.08** | 27.46 | "form" | 69.5 |
| "people" | **11.08** | 18.94 | "people" | 71.1 |
| "look" | **11.37** | 32.41 | "look" | 67.5 |
| "find" | **15.88** | 32.87 | "find" | 100 |
| "it" | **7.54** | 29.11 | "it" | 100 |
| "please" | **17.46** | 23.11 | "please" | 100 |
| "call" | 13.93 | **13.25** | "call" | 97.6 |
| "stella" | **14.45** | 15.77 | "stella" | 82.4 |
| Average | **14.43** | 24.77 | | |

(b) Tongue-twister words

| $w_B$ | $\|w_{B\to A} - w_B\|$ | $\min\limits_{w\in s_A}\|w - w_B\|$ | $\mathrm{mean}_{w\in s_A}\|w - w_B\|$ | $\arg\min\limits_{w\in s_A}\|w - w_B\|$ | Likelihood(%) |
|---|---|---|---|---|---|
| "please" | 17.46 | 23.11 | 53.93 | "please" | 100 |
| "things" | 21.59 | 19.87 | 50.95 | "things" | 81.8 |
| "these" | 25.22 | 21.26 | 82.00 | "these" | 55.6 |
| "cheese" | 21.98 | 20.34 | 51.19 | "cheese" | 100 |
| "three" | 24.39 | 22.13 | 61.07 | "three" | 84.1 |
| "mix" | 16.19 | 16.61 | 55.77 | "mix" | 100 |
| "six" | 20.43 | 16.75 | 66.85 | "six" | 100 |
| "thick" | 21.21 | 18.38 | 50.16 | "thick" | 92.3 |
| "kids" | 16.98 | 16.65 | 48.78 | "kids" | 100 |
| "is" | 16.84 | 16.25 | 70.88 | "is" | 78.0 |
| Average | 20.23 | 19.14 | 59.16 | | |

Table 5: **Verification of the reprojection property:** To verify the reprojection property that Eq. 7 (main paper) describes, we randomly sample 2 sets of words (Normal and Tongue-twister) spoken by 2 speakers (A and B) in VCTK dataset. For the autoencoder trained by A, all the words spoken by B are out-of-sample data. Then for each word $w_B$, we generate $w_{B\to A}$ which is the projection to A's subspace. We need to verify 2 properties - (i) The output $w_{B\to A}$ lies in A's subspace; (ii) The autoencoder minimizes the input-output error. To verify (i), we train a speaker classification network on speaker A and B, and predict the speaker of $w_{B\to A}$. We report the softmax output to show how much likely $w_{B\to A}$ is to be classified as A's speech (Likelihood in the table). To verify (ii), we calculate (1) distance from $w_B$ to $w_{B\to A}$; (2) minimum distance from $w_B$ to any sampled words by A. For normal words, the input $w_B$ is much closer (sometimes as close as) to the projection $w_{B\to A}$ than any other sampled point in A's subspace. Furthermore, such minimum is reached when the content is kept the same. For tongue-twister words, which are more confusable, we additionally calculate the mean distance from A's subspace to $w_B$. Distance from $w_B$ to $w_{B\to A}$ is close to the minimum, and much smaller than the mean. This empirically suggests that nonlinear autoencoder behave similar to their linear counterparts (i.e., they approximately minimize the reconstruction error of the out-of-sample input).

| VCTK (Veaux et al., 2016) | Voice Similarity (SCA / %) | Content Consistency (MCD) |
|---|---|---|
| **Chou et al.** (Chou et al., 2019) | 57.0 | 491.1 |
| **Auto-VC** (Qian et al., 2019) | | |
| with WaveNet | 98.5 | **408.8** |
| without WaveNet | 96.5 | **408.8** |
| **Ours** | | |
| with WaveNet | **99.6** | 420.3 |
| without WaveNet | **97.0** | 420.3 |

Table 6: **Ablation Analysis of WaveNet:** We replace the WaveNet vocoder with Griffin-Lim traditional vocoder, and compare our method with two zero-shot methods: (1) AutoVC without Wavenet, and (2) Chou et al. (Chou et al., 2019). We measure Voice Similarity by speaker-classification accuracy (SCA) criterion, where higher is better. The results show our approach without WaveNet still outperform other zero-shot approaches not using WaveNet. We also report MCD same as Table 2 in main paper as MCD is only related to Freq. features. For reference, we also list the results of "with Wavenet" experiments.

We perform new experiments that fine-tune a pretrained Auto-VC for each specific speaker (exemplar training) but still keeps the pre-trained embedding, and find it preforms similarly to the multi-speaker fine-tuning in Table 3 (main paper). Our outputs are still preferred $91.7\%$ times over it.

## D.5 Ablation Analysis of WaveNet

We adopt a neural-net vocoder to convert Mel-spectrograms back to raw audio signal. We prefer WaveNet (Oord et al., 2016) to traditional vocoders like Griffin-Lim(Griffin & Lim, 1984), since Wavenet is one of the state-of-the-art methods. The existence of WaveNet in our structure makes our method end-to-end trainable and can safely be considered a part of exemplar autoencoder (no special adaptation required). While Auto-VC(Qian et al., 2019) and VQ-VAE(van den Oord et al., 2017) also use WaveNet, we substantially outperform them on Celeb-Audio dataset.

Here we add the ablation analysis of WaveNet in Table 6. We remove the WaveNet vocoder from our approach and AutoVC, and replace it with Griffin-Lim. We also compare with Chou et al. (Chou et al., 2019), which does not use neural-net vocoders. The results show our model can also generate reasonable outputs with traditional vocoders, and yet outperform other zero-shot approaches not using WaveNet.

## D.6 Ablation Analysis of Video Synthesis

In main paper, we propose "exemplar autoencoders" and prove the effectiveness of jointly training a audio-to-audio-video pipeline based on a pre-trained audio model. To further verify the helpfulness of audio bottleneck features and finetuning from audio model in video synthesis, we conduct two ablation experiments on audio-to-video translation, and compare with ours.

(a) Train only the audio-to-video translator. (First Baseline in Table 7)

(b) Jointly train audio decoder and video decoder from scratch. (Second Baseline in Table 7)

(c) First train an autoencoder of audio, then train video decoder while finetuning the audio part. (Ours)

From the results in Table 7, ours outperforms (b), which indicates the effectiveness of finetuning from a pre-trained audio model. And (b) also outperforms (a). This implies the effectiveness of audio bottleneck features.

## D.7 Details of Human Studies

All the users of AMT were chosen to have Master Qualification (HIT approval rate more than $98\%$ for more than $1,000$ HITs). We also restricted the users to be from United States to ensure English-speaking audience.

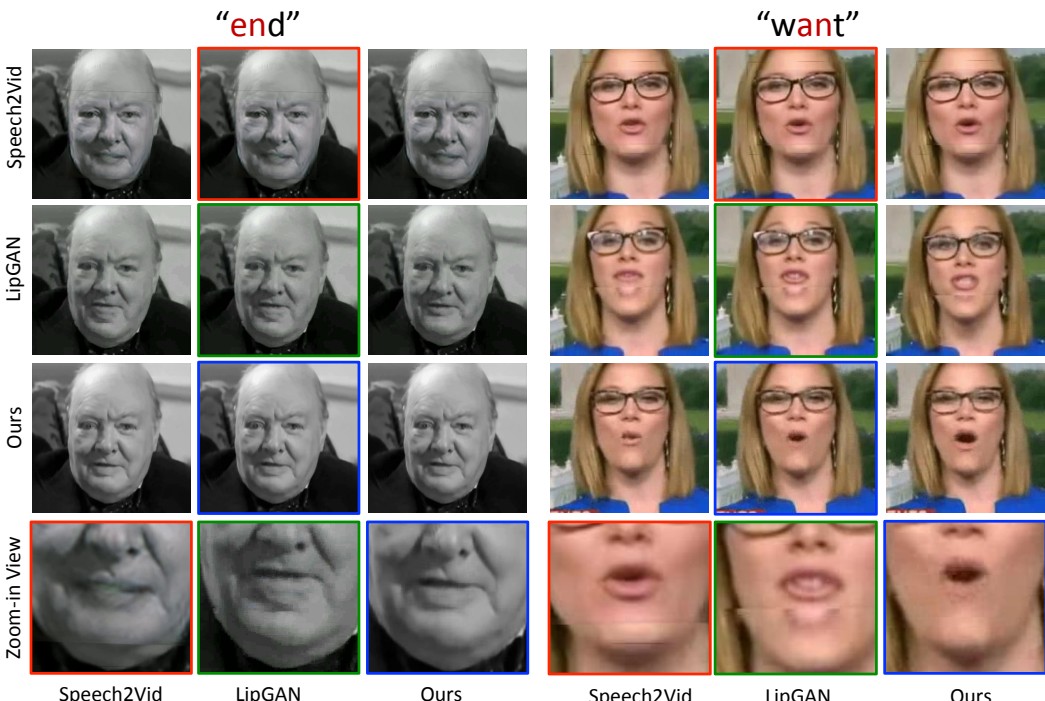

Figure 6: **Visual Comparisons of Audio-to-Video Synthesis:** We contrast our approach with Speech2vid (Chung et al., 2017) (first row) and LipGAN (KR et al., 2019) (second row) using their publicly available codes. While Speech2Vid synthesize new sequences by morphing mouth shapes, LipGAN pastes modified lip region on original videos. This leads to artifacts (see zoom-in views at bottom row) when morphed mouth shapes are very different from the original or in case of dynamic facial movements. Our approach (third row), however, generates full face and does not have these artifacts.

| VoxCeleb | Finetune | Audio Decoder | MSE↓ | PSNR↑ |
|---|---|---|---|---|
| Baselines | ✗ | ✗ | $77.844 \pm 15.612$ | $29.304 \pm 0.856$ |
| | ✗ | ✓ | $77.139 \pm 12.577$ | $29.315 \pm 0.701$ |
| Ours | ✓ | ✓ | $\mathbf{76.401 \pm 12.590}$ | $\mathbf{29.616 \pm 0.963}$ |

Table 7: **Ablation Analysis of Video Synthesis:** To verify the effectiveness of a pre-trained audio model and audio decoder, we construct 2 extra baselines: a. Train only the audio-to-video translator (1st Baseline). b. Jointly train audio decoder and video decoder from scratch (2nd Baseline). And we compare these 2 baselines with our method: First train an autoencoder of audio, then train video decoder while finetuning the audio part. From the results, we can see the performance clearly drops without the help of finetuning and audio decoder.

