# OpenReview forum: "Unsupervised Audiovisual Synthesis via Exemplar Autoencoders"
_ICLR.cc/2021/Conference — ICLR 2021 Poster_

### Official Review · AnonReviewer4 · 2020-10-29
**Recommendation Natural**

**Rating:** 6
**Confidence:** 3

**Review:**

This paper covers a very interesting topic and method to convert any input speech to many audiovisual syntheses via exemplar autoencoders. The manuscript is well written and presented. It is easy to follow the concept. However, there are a few major concerns.

Pros: This approach is novel. The presented approach is unsupervised, which makes it more practical. This approach required only a small amount of data to train the exemplar autoencoders when learning specialized models tailored to particular target data.

cons:
1- After listening to the provided sample demo files. I think this approach is still in an immature status. The generated output speech is highly distorted, without knowing the input speech, it is hard to understand the generated speech. Hence, more work is required to improve on the audio decoder part.
2- It is not mentioned if the data used for training is all speech of native English speakers or not.
3- Any reasons authors did not use the x-vector or i-vectors, which are proper for the speaker characterization, and instead used mel features?

---

> ### Author Response · Authors · 2020-11-18
> **Good questions. We address below.**
>
> We thank the reviewer for bringing up these points:
>
> **Results could be better**
> We agree that our current results can still be improved. Our method is based on manipulations of Mel-spectrograms. This makes the quality of our generation results heavily dependent on the wavenet vocoders, which reconstruct raw audio signals from Mel-spectrograms. From our observation, the transformation between Mel-spectrograms and audio signals is not lossless, especially for those in-the-wild audio samples. We believe our results will benefit from a more advanced vocoder, as well as a model structure with raw audio input. Future work in these directions using in-the-wild dataset will help address the challenges.
>
> **Are models trained with native English speakers?**
> The standard VCTK dataset only contains native English speakers with various accents. Our CelebAudio dataset contains both native (e.g. John Oliver) and non-native (e.g. Takeo Kanade) English speakers. We have clarified this in Sec 4.1 in our revised edition and provided dataset details in the appendix.
>
> **Why not use i-vectors/x-vectors instead of Mel spectrograms?**
> I-vectors [3] and x-vectors [4] are learned features for speaker classification (which is a different task than speech synthesis), while Mel spectrograms are non-learned Fourier features. The heart of our approach is learning representations on the target clip of interest, rather than a large training corpus with many styles. Using pre-trained representations may limit our ability to capture target-specific styles, but specializing them to the target could potentially help. For simplicity, we use non-learned Fourier representations used by other voice conversion approaches [1,2].
>
> [1] Qian, Kaizhi, et al. "Autovc: Zero-shot voice style transfer with only autoencoder loss." arXiv preprint arXiv:1905.05879 (2019).
>
> [2] Chou, Ju-chieh, Cheng-chieh Yeh, and Hung-yi Lee. "One-shot voice conversion by separating speaker and content representations with instance normalization." arXiv preprint arXiv:1904.05742 (2019).
>
> [3] Ibrahim, Noor Salwani, and Dzati Athiar Ramli. "I-vector extraction for speaker recognition based on dimensionality reduction." Procedia Computer Science 126 (2018): 1534-1540.
>
> [4] Snyder, David, et al. "Deep Neural Network Embeddings for Text-Independent Speaker Verification." Interspeech. 2017.

---

### Official Review · AnonReviewer1 · 2020-10-29
**Interesting paper covering a lot of ground; exposition could be tightened/clarified?**

**Rating:** 6
**Confidence:** 3

**Review:**

Interesting paper covering a lot of ground; exposition could be tightened/clarified?

Super interesting topic; strong references to prior work (I add a reference at the very end that I think you'd be interested in, Hani Yehia et al. (2002).)

I found the paper slightly verbose -- I'm not sure i fully "got" the central concept of Exemplar Autoencoders. Looking at Section 3, with that in the title, I was waiting to find the canonical definition of E.A.'s, but don't quite see it -- it seems somewhat unclear given the many variants of auto-encoders discussed. The discussion in Section 2, and e.g. Figure 2c, help clarify what the author's mean by EA's. What I'm not fully sure about is, are EA's basically what I would call Auto-encoders? Fig 3c is labeled "Exemplar Autoencoder", but then the caption for 3c mentions "Non-linear autoencoder", which is confusing. The caption starts with "Figure 3: Our insights for Exemplar Autoencoders, " leading me to think that actually all three of these sub-figures are variants of Exemplar Autoencoders. Maybe they could clarify their terminology or re-read the draft from the perspective of someone who is not as "close" to the work as they are.

The contrast with zero-shot conversion is very interesting -- but here too I feel I am somewhat missing the explanation of the essence of these zero-shot methods -- though it is possible most readers do not need any additional explanation. In particular, when I see Fig 2a, I'm wondering how the content and speaker embedding vectors are trained (and on what data), and the text doesn't quite clarify that for me.

More specific comments follow:

"(a) Zero-shot conversion that learns a generic low-dimensional embedding from a training set that _are_ designed ...": "are" --> "is".

" In supp material, ..." --> "In supplementary material, ..."

Define "retargeting"?

Table 2: Use up & down errors, as you did in a later table, to indicate whether higher/lower is better.

No conclusion? It seems you have one, but it's "Discussion", as a subsection of the last experiments section.

The references seem very comprehensive, but I urge you to look at, and cite, Yehia et al. 2002,

Yehia, Hani Camille & Kuratate, Takaaki & Vatikiotis-Bateson, Eric. (2002). Linking facial animation, head motion and speech acoustics. Journal of Phonetics. 30. 555-568. 10.1006/jpho.2002.0165.

as a seminal study in this area.

---

> ### Author Response · Authors · 2020-11-18
> **Helpful Suggestions. We have modified the manuscript.**
>
> We thank the reviewer for their careful review and pointer to the related work. We address specific concerns below.
>
> **What are Exemplar Autoencoders?**
> We apologize for the confusion and have rewritten the intro paragraph in Sec.3 and the caption of Fig.3 to succinctly explain our approach: exemplar autoencoders are simply autoencoders trained on a single clip containing a single speech style. We use the term exemplar to contrast them with autoencoders trained on large-scale datasets spanning many clips and many styles. Training autoencoders on a single style allows them to act as translation engines that preserve out-of-sample content (words) while transferring onto the (single) target style.
>
> **Comparison to Zero-Shot Approaches. How do they work?**
> Prior work (such as Auto-VC [1]) trains a model for speech conversion using a large corpus of speeches from many speakers. Once trained, this model can be used to synthesize the voice of a new person who was not seen during training by requiring a small sample of speech from the target person at test-time. This small sample is processed by a style encoder (trained on a large corpus) that returns a style embedding. We denote this with the “input style from target speaker” text in Fig-2a. In some sense, our approach dispenses with the large corpus of training data altogether and uses only this small target speech sample (to train an exemplar autoencoder).
>
> **What does retargeting mean?**
> This term has been used in the computer graphics literature for motion retargeting or video retargeting, which often implies resynthesizing media content with a new identity. We will clarify.
>
> [1] Qian, Kaizhi, et al. "Autovc: Zero-shot voice style transfer with only autoencoder loss." arXiv preprint arXiv:1905.05879 (2019).

---

### Official Review · AnonReviewer3 · 2020-10-29
**Well written paper with interesting results. Worth publibication.**

**Rating:** 9
**Confidence:** 4

**Review:**

In this paper, the authors propose a generic system for performing one-shot audiovisual synthesis from only one small sample. The results are impressive for in-the-wild speech synthesis and their approach could have a broader impact in the community.

Strengths:
 + One shot audiovisual synthesis for a target speaker.
 + The publication of a new dataset for AV synthesis evaluation.
 + Comprehensive analysis

Weaknesses:
 - No theoretical novelty. It seems much of the benefits of the approach comes from the extra data and training procedure.

Other comments:

In Sec 3.2. Autoencoders as projection operators, the authors here make it sound that they are the first ones that noted that autoencoders can capture the data-generating distribution.

---

> ### Author Response · Authors · 2020-11-18
> **Valuable feedback. We agree.**
>
> We thank the reviewer for valuable feedback.
>
> **Authors imply that they are the first to note that autoencoders capture the data-generating distribution.**
> The reviewer is quite right that we should not make this claim. We reframed Sec 3.2 to highlight “autoencoders for style transfer”, rather than “autoencoders as projection operators”. We have added previous work in “Sec 2 Autoencoders” that explicitly points out this projection property (e.g., denoising autoencoders [1] that project noisy inputs onto the data manifold). To our knowledge, autoencoders have been primarily applied to tasks such as dimensionality reduction and feature learning [2], while we repurpose their projection property for style transfer (by re-stylizing out-of-sample content onto the style of the target). We believe this is a novel use of the (well-known) ability of autoencoders to project onto the data manifold.
>
> [1] Pascal Vincent, Hugo Larochelle, Yoshua Bengio, and Pierre-Antoine Manzagol. Extracting and composing robust features with denoising autoencoders. InProceedings of the 25th international conference on Machine learning, pp. 1096–1103, 2008
>
> [2] Yoshua Bengio, Ian Goodfellow, and Aaron Courville. Deep learning, volume 1.  MIT press Massachusetts, USA: 2017.

---

### Decision · Program_Chairs · 2021-01-07
**Final Decision**

**Decision:**

Accept (Poster)

**Comment:**

This paper proposes and investigates an approach for audiovisual synthesis based on the so-called exemplar autoencoders.  The proposed approach is shown to be able to convert an audio input to audiovisual outputs using only very small amount of training data.  All reviewers consider the paper interesting with a lot of potentials in a variety of applications and appreciate the novelty of the work in this domain.  But there are also concerns on the technical presentation and the quality of the samples in the demo.  The authors addressed most of the concerns in the rebuttal but agreed that the quality of the results still had room for further improvements.  Overall, the work presented is interesting. The paper can be accepted.